# A High-Performance Antibacterial Nanostructured ZnO Microfluidic Device for Controlled Bacterial Lysis and DNA Release

**DOI:** 10.3390/antibiotics12081276

**Published:** 2023-08-02

**Authors:** Yvonni Xesfyngi, Maria Georgoutsou-Spyridonos, Abinash Tripathy, Athanasios Milionis, Dimos Poulikakos, Dimitrios C. Mastellos, Angeliki Tserepi

**Affiliations:** 1Institute of Nanoscience and Nanotechnology, National Center for Scientific Research (NCSR) “Demokritos”, Patr. Gregoriou E’ and 27 Neapoleos Str., 15341 Aghia Paraskevi, Greece; yvonnicx@gmail.com (Y.X.); maria.georgoutsou.spyridonos@gmail.com (M.G.-S.); 2Laboratory of Thermodynamics in Emerging Technologies, Department of Mechanical and Process Engineering, ETH Zurich, 8092 Zurich, Switzerland; atripathy8@gmail.com (A.T.); athanasios.milionis@ltnt.iet.mavt.ethz.ch (A.M.); dpoulikakos@ethz.ch (D.P.); 3Institute of Nuclear & Radiological Sciences and Technology, Energy & Safety, National Center for Scientific Research (NCSR) “Demokritos”, Patr. Gregoriou E’ and 27 Neapoleos Str., 15341 Aghia Paraskevi, Greece; mastellos@rrp.demokritos.gr

**Keywords:** nanostructured ZnO, antibacterial surfaces, bacterial lysis, static microfluidic chip, bacterial DNA, reactive oxygen species

## Abstract

In this work, the antibacterial properties of nanostructured zinc oxide (ZnO) surfaces are explored by incorporating them as walls in a simple-to-fabricate microchannel device. Bacterial cell lysis is demonstrated and quantified in such a device, which functions due to the action of its nanostructured ZnO surfaces in contact with the working fluid. To shed light on the mechanism responsible for lysis, *E. coli* bacteria were incubated in zinc and nanostructured ZnO substrates, as well as the here-investigated ZnO-based microfluidic devices. The unprecedented killing efficiency of *E. coli* in nanostructured ZnO microchannels, effective after a 15 min incubation, paves the way for the implementation of such microfluidic chips in the disinfection of bacteria-containing solutions. In addition, the DNA release was confirmed by off-chip PCR and UV absorption measurements. The results indicate that the present nanostructured ZnO-based microfluidic chip can, under light, achieve partial inactivation of the released bacterial DNA via reactive oxygen species-mediated oxidative damage. The present device concept can find broader applications in cases where the presence of DNA in a sample is not desirable. Furthermore, the present microchannel device enables, in the dark, efficient release of bacterial DNA for downstream genomic DNA analysis. The demonstrated potential of this antibacterial device for tailored dual functionality in light/dark conditions is the main novel contribution of the present work.

## 1. Introduction

In recent years, microfluidics has exhibited remarkable potential in molecular diagnostics as an enabling technology that conceptualizes chemical and biological analysis by re-creating a research laboratory on a miniaturized chip-scale device, thus leading to its wide use in molecular biology, clinical medicine, and biomedical science [1,2,3]. Microfluidics has also gained popularity in cell biology research due to its ability to precisely manipulate cells and recapitulate cellular environments [4], its precision and reproducibility through standardization, and ease of integration in prototyping platforms. For a wide range of applications, such as point-of-care diagnostics for rapid detection of pathogens [5,6,7], genetic screening of biological samples [8], and food safety [9], cell lysis is required as an operation step for gene analysis, followed by nucleic acid purification, amplification, and detection [10]. Cell lysis occurs when the cell membrane’s outer layer is broken down in order to liberate intracellular components that hold important genetic information (DNA, RNA). Microfluidics hold a potential for rapid and convenient on-chip cell lysis when compared to conventional methods.

Microfluidic-based cell lysis constitutes a simple and promising alternative to conventional lab methodology that has the potential to revolutionize diagnostics, due to shorter turnaround times, high efficiency, decreased reagent consumption, and lower cost due to its scale-up capability [9,11]. Chemical, mechanical, thermal, electrical, and laser lysis are the most commonly used methods for cell lysis, demonstrated in various microfluidic devices for genomic DNA analysis [12,13]. From the aforementioned on-chip lysis methods, the reagent-free ones are preferable due to their broader applicability to all types of cells (bacterial, human, fungal) and their independence from the subsequent removal of unwanted chemical substances. However, electrical and laser-based lysis methods are more technically demanding and expensive. Finally, mechanical lysis in the majority of microfluidic devices requires the creation of complex geometrical structures/obstacles on the microreactor walls [14,15,16] to enhance the shear forces applied on the cell membrane, thus increasing the device’s fabrication complexity and cost. Therefore, the microfluidics community is still in need of a simple-to-fabricate, low-cost, and easy-to-use microfluidic device for reagent-free bacterial lysis. 

The development of antibacterial materials with nanostructured features has shown promising results for reducing the adhesion on surfaces [17,18] or even killing bacteria due to the fact that cells respond to nanostructures of specific topography by deforming or extended stretching of their membrane [19,20,21], resulting to cell piercing or rupture [22,23,24,25,26]. The most well-known materials with active antibacterial properties are nanosized metal oxide compounds and nanostructures thereof (TiO_2_, ZnO, CuO) which are exploited as antimicrobial surfaces [19,27,28], as well as in microfluidic and highly efficient analytical platforms [29]. In addition to their nanostructured surface properties, the formation of reactive oxygen species (ROS) strengthens the antibacterial activity of metal oxide-based materials [30,31]. ROS, owing to their high oxidative potential, can cause biomolecule damage, such as lipid or protein peroxidation, DNA or RNA damage [32], and membrane structure destruction [33,34].

In particular, the antibacterial activity of ZnO-based nanomaterials has sparked widespread attention due to their effect on biological species and their appealing physicochemical properties [35,36]. The most well-known type of ZnO with high antibacterial potential is zinc oxide nanoparticles [36,37]. Moreover, many nanosized ZnO structures (nanowires, nanorods, and nanoflowers) have improved antibacterial properties and have been implemented in on-chip applications [38]. Two main mechanisms of action have been proposed as responsible for the antibacterial properties of ZnO (as well as of other metal-oxide materials as described in the previous paragraph), namely, the production of reactive oxygen species (ROS) and the loss of cellular integrity and membrane rupture after contact of ZnO materials with the cell wall [39]. Both of these mechanisms lead to cell lysis and release of bacterial DNA, as has been already demonstrated in a few cases [19,29]. Therefore, the antibacterial properties of nanostructured ZnO can also be exploited as promoting lysis of bacterial cells. To fabricate ZnO nanostructures, commonly used techniques are physical vapor deposition [40], chemical bath deposition [41], electrochemical-thermal approach [42], sol–gel [43], and pulsed laser deposition [44]. Many of these techniques necessitate sophisticated and expensive equipment in a controlled environment. On the contrary, a hot water treatment (HWT) process provides a simple and eco-friendly approach for the fabrication of ZnO nanostructures [45,46,47] with antibacterial activity [48].

Despite the remarkable capabilities in biosensing and biological separation demonstrated by virtue of the integration of ZnO materials with microfluidics [38], the implementation of ZnO-integrated microfluidics for cell lysis is less explored. Specifically, ZnO nanowire-integrated microfluidic devices for cell lysis have been demonstrated in a single report [29] for mechanically lysing a variety of immortalized human cells flowing through the microchannel. However, the fabrication of such devices entailed photolithography and other clean-room processing, as well as alignment before sealing, rendering the device fabrication quite sophisticated and costly. Herein, a low-cost microfluidic device with a nanostructured ZnO surface, fabricated through a facile, scalable, and environmentally benign (water-based) strategy, is reported, where *E. coli* bacteria were lysed with unprecedented killing rates even after short incubation at static conditions. In addition, the released DNA was evaluated off-chip to confirm efficient bacteria lysis in the device. The obtained results shed light on the mechanisms responsible for *E. coli* bacteria killing in the presence of nanostructured ZnO surfaces. This microfluidic chip for bacterial lysis does not require complicated design, chemical reagents, or power consumption, which makes it an attractive on-chip lysis device, easily integrable with other microfluidic modules necessary for genetic analysis and point-of-care DNA-based diagnostics. Furthermore, this device can be implemented for efficient inactivation of undesirable DNA, where necessary, replacing exposure to UV radiation. 

## 2. Results and Discussion

### 2.1. Surface Morphology 

Figure 1 shows the morphology of ZnO nanostructures formed on a zinc substrate after 24 h immersion in a hot water bath (95 °C), according to the method described in detail in [48]. Briefly, upon immersion of zinc substrates in hot water, ZnO nanocrystals are formed as a result of the fast adsorption and break-up of water molecules on the zinc surface, serving as seeds for further growth to ZnO nanowires (Figure 1a). ZnO nanorods with hexagonal tips are grown (Figure 1b) as a result of nanocrystal self-arrangement and preferential diffusion of Zn ions to the tips of ZnO nanostructures at the initial stages of nanorod formation, or redeposition onto the surface of dissolved ZnO at long immersion times. After an extended (24 h) hot water treatment, as in this work, nanorods several μm in length and with diameters in the range 130–400 nm (Figure 1b) are formed. The formation of metal oxide nanostructures on metal surfaces immersed in hot water has been demonstrated in many reports in the literature, where a wide spectrum of applications has been presented [45,46]. In this work, ZnO nanostructures will be evaluated as antibacterial surfaces by direct contact or indirect bacteria killing and as microchannel surfaces for subsequent DNA release. 

### 2.2. Antibacterial Activity of Nanotextured ZnO Substrates and Microchannel Walls

The antibacterial activity of the nanostructured ZnO surfaces was evaluated in comparison with flat Zn and PMMA substrates (of two sizes, 1 × 1 or 1.5 × 1.5 cm^2^) using the standard bacterial colony counting method. In more detail, the standard bacterial colony counting method is used to quantitate the concentration of bacteria in a test sample by applying serial dilutions of the bacterial suspension on standard bacteriology (agar) plates and incubating the plates overnight under optimal culture conditions for the particular strain in question. Bacterial concentration (expressed as CFU/mL) is determined by counting the number of colonies grown out of a prespecified diluted sample (colony-forming units, CFU) and adjusting for the dilution factor and starting volume [49]. After incubation of an initial *E. coli* concentration (~10^6^ CFU/mL or ~10^8^ CFU/mL) suspended in 0.9% NaCl saline solution with the surfaces at 37 °C, bacterial suspensions were collected from the top of the test samples, and the supernatant was inoculated to agar plates in triplicates for each sample to enumerate the live bacteria. Figure 2 shows the number of viable cells in the respective supernatant of the samples. After incubation, no inhibition of the bacterial viability was observed on the PMMA substrate, which was taken as a control in our experiments. In fact, the viability of *E. coli* on PMMA did not differ (within error) from the time-zero (TZ) bacterial concentration, implying that the bacteria were kept alive on flat PMMA surfaces. In contrast, much-less viable bacteria were found on top of the flat zinc and nanostructured ZnO surfaces (especially in the case of larger (1.5 × 1.5 cm^2^) surface area substrates), the concentration of which was four orders of magnitude lower than on the flat PMMA (Figure 2b), even after a short incubation time (15 min). This incubation time was reduced from 1 h to avoid using extremely high bacteria concentrations (higher than 10^7^ CFU/mL) coming into contact with large surfaces (1.5 × 1.5 cm^2^). In any case, the bacteria killing rates (Table 1) are reported per unit area and time (CFU/cm^2^ h), allowing direct comparison between the different surfaces. 

Although in both cases (small and large surfaces), the concentration of surviving bacteria observed after incubation on Zn and nanostructured ZnO surfaces show a similar difference (around 40% decrease on ZnO with respect to Zn), this is not reflected in a similar way on the killing rates of *E. coli* on the two substrates. Indeed, the killing rate on 1 × 1 cm^2^ ZnO surfaces was found to be 40% higher compared to the respective Zn surfaces (see Table 1), while on the contrary, similar killing rates for *E. coli* were observed in the case of larger surface area substrates due to the very high value of the concentration of surviving bacteria on control surfaces (~10^7^ CFU/mL). The aforementioned results indicate that the ZnO nanostructures contribute (by up to 40%) to direct bacteria killing if sufficient time is given for their interaction with the bacteria (~1 h). On the larger surface area (1.5 × 1.5 cm^2^) Zn and ZnO substrates, indirect bacteria killing is so intense at short incubation times (15 min) that the ZnO surface nanostructures do not have a noticeable effect on the killing outcome, possibly due to reduced bacteria-surface interaction time.

However, the implementation of such large surfaces enables the antibacterial (indirect killing-based) activity of Zn and ZnO to be observed even at very short incubation times (15 min, Figure 2b), rendering promising their use as microchannel walls for fast (indirect) bacteria killing in solutions injected in the microchannel.

Following this observation, a meandering microchannel was fabricated with nanotextured ZnO walls (with surface area of 1.54 cm^2^) and was reversibly sealed with a PDMS cover (see Figure 3a) through which an inlet and outlet were opened for bacteria solution injection and collection, respectively, after incubation. A similar microchannel was fabricated on a PMMA substrate and *E. coli* were incubated in both microchannels. After a short incubation time (15 min), the bacteria solution was collected from the microchambers and was inoculated in a similar manner to the substrates. The viability of *E. coli* in the PMMA and ZnO microchannels is shown in Figure 3b. In comparison to the PMMA microchannel (control), the killing efficiency of the nanostructured ZnO microchannel for *E. coli* was 96% after 15 min incubation.

From data collected as in Figure 2 and Figure 3, the killing rates for *E. coli* on the aforementioned substrates and microchannel surfaces are shown on Table 1. The killing rate for *E. coli* appears to be significantly higher (10^7^ CFU/cm^2^ h) on larger (1.5 × 1.5 cm^2^) Zn and ZnO surfaces, demonstrating strong antibacterial activity against *E. coli*, whereas it was lower (by two orders of magnitude) on smaller (1 × 1 cm^2^) surfaces. On similar (1 × 1 cm^2^) nanostructured ZnO substrates, smaller killing rates by one order of magnitude have been reported [48]. The here-reported higher killing rates can be attributed to the rich ZnO topography, as indicated by the thin, sharp, and tall ZnO nanorods in Figure 1. The killing rate for *E. coli* observed in the present ZnO microchannels (Table 1) is in between the values observed for the ZnO substrates, indicating a monotonic dependence on the ZnO surface area. In fact, an exponential dependence of the killing rate on the surface area, indicated by the results, will be claimed in the following paragraph. Two-tailed *t*-tests were performed for comparing statistically the killing rates (six measurements per case) among the different surfaces. In all cases, with the exception of the 1.5 × 1.5 cm^2^ Zn and ZnO surfaces, the calculated t-values were found greater than the critical t-values, meaning that the experimentally determined killing rates are significantly different. 

The antibacterial performance of these surfaces can be attributed to multiple mechanisms, such as indirect killing of the bacteria due to reactive oxygen species (ROS) formed in both Zn and ZnO substrates under light conditions, as well as direct (mechanical) killing due to the contact of bacteria with the nanorods on nanostructured ZnO substrates [47,48]. The latter mechanism contributes, to a large extent (approximately 30% as estimated by comparing the killing rates on nanostructured ZnO and flat Zn substrates (Table 1)), to the total killing rate, though only in the case of small area substrates (1 × 1 cm^2^) and sufficiently long bacteria-surface interaction (~1 h). In the case of large surface area substrates (1.5 × 1.5 cm^2^) and short bacteria-surface interaction (15 min), bacteria killing is dominated by indirect killing as a result of ROS formation during the incubation of bacteria with Zn and ZnO surfaces, which is to be discussed in Section 2.4. An exponential dependence of *E. coli* killing rates on the ZnO surface area, supported by the results in Table 1, is justified via analogy to exponential models describing the bactericidal activity of antibiotics [50]. Thus, larger ZnO surfaces are expected to result in remarkably enhanced bacterial killing rates, as demonstrated herein. Detailed study of the dependence of the bacterial killing rate on the ZnO surface area is beyond the scope of the present work. Nevertheless, the pronounced killing rate for *E. coli* (~10^6^ CFU/cm^2^ h) observed in the present ZnO microchannels, even after a 15 min incubation of the bacteria, paves the way for the implementation of such surfaces in the disinfection of bacteria-containing solutions injected in such easy-to-fabricate microchannels. Furthermore, the same microchannel surfaces will be assessed in the next section for enabling efficient bacteria lysis and DNA release for subsequent DNA analysis. 

### 2.3. Bacterial DNA Release Following Bacterial Cells Exposure to ZnO Surfaces 

To assess the potential of nanostructured ZnO microchannel surfaces to enable DNA release after bacteria killing, the bacteria solution was collected at the outlet of the microchannel after incubation in the microchannel at static conditions for 15 min. Static conditions were preferred over flow conditions for the bacteria residence in the microchannel, as the latter would result in less bacteria interacting with the nanostructured substrate, due to the parabolic velocity profile. This, in turn, would reduce the DNA release especially in dark conditions, where direct (mechanical) bacteria killing contributes significantly to DNA release, as will be shown below. Static conditions are also preferred so that the released DNA is concentrated with time in the microchannel before the DNA sample proceeds downstream for further analysis. The released bacterial DNA was detected by means of UV absorption (Nanodrop), as well as gel electrophoresis, following DNA amplification based on polymerase chain reaction (PCR). Specifically, the ybbw gene of *E. coli* [51] was amplified by PCR, and the amplification results are shown in Figure 4a for samples collected from a nanostructured ZnO and a flat PMMA (control) microchannel. Despite the low yield of PCR product due to the use of unpurified DNA (whole bacterial lysate), amplified DNA signal was observed in all cases. Apparently, bacteria killing in the microchannel results in DNA release. Surprisingly, the gel images in Figure 4a (for 1:60 sample dilution) indicate that the released bacterial DNA is higher in the PMMA microchannel (by a factor of 3, as estimated by comparing the fluorescence intensity of the corresponding bands using ImageJ) than in the ZnO microchannel, despite the much higher bacteria killing in the ZnO microchannel (Figure 3b). To shed light on this result, purified DNA was injected in the microchannels and was collected after 15 min. Figure 4b shows PCR products obtained from the collected DNA and demonstrates a quantity of amplified DNA from the ZnO microchannels smaller by a factor of 1.5 compared to that from the PMMA microchannel (for 1:60 sample dilution) as estimated by comparing the fluorescence intensity of the corresponding bands using ImageJ, indicating partial inactivation of original DNA in the ZnO microchannel. 

In agreement with the PCR results, absorption measurements (on NanoDrop spectrophotometer) shown in Table 2 assessed the released DNA, indicating similar trends. The concentration of DNA from bacteria lysed in the nanostructured ZnO microchannel was three times lower than in the flat PMMA microchannel. In the latter, the purified DNA concentration was maintained the same after a 15 min residence; thus, DNA does not suffer any damage in the PMMA microchannel. On the contrary, the purified DNA concentration was reduced to nearly 50% in the nanostructured ZnO microchannel, thus revealing partial DNA inactivation. Presumably, generation of ROS as a result of the photocatalytic properties of the ZnO substrate is responsible for the observed behavior. Destruction of the bacteria membrane together with the oxygen radical-mediated inactivation of the released DNA has been reported in the literature as a result of the bactericidal action of metal oxide nanoparticles [52,53]. ZnO surfaces, similarly to ZnO or silver nanoparticles, cause inactivation of DNA, presumably due to the generation of ROS near ZnO surfaces.

A question that also needs to be addressed concerns the strong amplified DNA signal from the PMMA microchannel (Figure 4a) in contrast to the very low bacteria killing observed in this microchannel (Figure 3b). It was assumed that the light conditions play a role in this, as will be demonstrated by monitoring released DNA by means of fluorescence. Ethidium bromide (EtBr), utilized routinely in gel electrophoresis, intercalates between base pairs of free DNA (ssDNA or dsDNA [54,55]) and, under UV light excitation, (526 nm) emits light at 605 nm to probe, in this case, DNA released from lysed *E. coli*. EtBr fluorescence is directly proportional to the concentration of intact DNA in the microfluidic channel as demonstrated by the linearity between fluorescence intensity and DNA in Appendix A. If the DNA sustains ROS-mediated oxidative damage, it would be expected not to incorporate EtBr as efficiently due to possible degradation/fragmentation or even due to a possible fluorescence quenching effect following the incorporation of EtBr to the modified DNA. An initial *E. coli* concentration of 10^8^ CFU/mL was injected in the microchannels and was incubated in static conditions for 15 min under light or dark conditions, and EtBr (0.4 μL/mL) was added at 10 μL of the solution collected at the microchannel outlet (after centrifugation to pellet the cell debris on the bottom, transferring the DNA supernatant to react with EtBr). Fluorescence measurements were performed in triplicate for each tested sample, and two independent experiments were performed, the average values of which are shown in Figure 5.

Comparison of fluorescence intensity (from released intact DNA) between light and dark conditions for the PMMA microchannel reveals that the presence of light induces bacteria lysis (presumably due to ROS formation, as will be shown in Figure 6) and thus DNA release in PMMA microchannels. This justifies the strong signal of amplified DNA from the PMMA microchannel (Figure 4a). In addition, comparison of fluorescence intensity (from released DNA) between light and dark conditions for the Zn microchannel reveals that the presence of light induces bacteria lysis (presumably due to ROS formation) and thus DNA release in the Zn microchannel, while no bacteria lysis (and thus DNA release) occurs in the dark. The latter is justified by the absence of direct (mechanical) as well as indirect (ROS-induced) bacteria killing, due to the absence of ROS production in the dark for flat Zn surfaces [48]. On the contrary, light influences the released DNA in the ZnO microchannel in an opposite way. Indeed, in the presence of light, a decrease by a factor of 1.5 in the released DNA is observed in the ZnO microchannel, indicating partial inactivation of the released DNA. The finding that the DNA concentration from lysed bacteria under light conditions is higher in PMMA microchannels than in nanostructured ZnO microchannels is verified via two independent methods (PCR amplification and EtBr fluorescence). Subsequently, in Section 2.4, this result will be attributed to partial oxidative stress-induced inactivation of the released DNA under light conditions in the ZnO microchannel, due to the highest ROS% production with respect to PMMA. Thus, dark conditions are recommended during operation of the present microdevice for bacterial lysis, if the released DNA is intended for downstream analysis, and thus should remain intact. In fact, Figure 5 indicates that in the dark, the released DNA in the nanotextured ZnO microchannel is several orders of magnitude greater than in the other microchannels (PMMA, Zn), where the released DNA is at the baseline (control well). 

Partial inactivation of DNA in the ZnO microchannel under light is justified by the well-known photocatalytic properties of ZnO that lead to the production of ROS as a result of various chemical reactions [56,57]. Can the released DNA in the PMMA microchannel be attributed to the same cause, i.e., the production of ROS? To shed light on this issue, the production of ROS in a variety of surfaces and under light/dark conditions is examined and discussed in Section 2.4. Nevertheless, the oxidative stress produced in environments containing ROS is well-known to have a strong effect on DNA, leading to its inactivation and damage [32,58]. This justifies the reduction (by a factor of 1.5) of DNA probed by fluorescence in the ZnO microchannel under light conditions (Figure 5) and agrees with the gel electrophoresis (Figure 4b) as well as the absorption measurements (Table 2) for the quantity of DNA from the ZnO microchannel. 

### 2.4. Reactive Oxygen Species (ROS)

In the previous section, the role of light in the release of intact DNA from lysed bacteria was demonstrated. In this section, measurements of relative ROS release (%), conducted as an additional method to point out the importance of ROS production under dark and light conditions, are shown for up to 5.5 h. Figure 6a shows that the nanostructured ZnO surface generated the most ROS, almost three-fold higher than the flat PMMA surface (over 0.5 h, the closest time to the residence time of bacteria in the microchannel). This enhanced ROS production contributes to the higher killing efficiency of the bacteria but also, at the same time, to a higher partial inactivation of the released DNA, thus leading, overall, to a smaller intact DNA concentration for the ZnO microchannel. The production of ROS in ZnO increased with time under both light (Figure 6a) and dark (Figure 6b) conditions. In presence of ZnO, the ROS generation was smaller in the dark than under light conditions, but still significant compared to PMMA surfaces (where ROS generation is minimal in the dark). Under light conditions, ROS were produced in the presence of PMMA (although much less compared to ZnO), and this justifies the overall DNA release results obtained in this work with PMMA surfaces, despite the low bacteria killing. These results clearly demonstrate the influence of light on indirect bacteria killing through the ROS-mediated oxidative damage of DNA and membrane structure destruction in the ZnO microchannel. Indirect bacteria killing due to ROS production under light occurs also in PMMA microchannels; however, the produced ROS concentration is not sufficiently high to cause inactivation of DNA released in PMMA microchannels. This is further supported by Figure 5, where the smaller ROS production in dark conditions for the nanostructured ZnO surface leads indeed to a (1.5-fold) higher intact DNA concentration compared to light conditions. Therefore, bacteria lysis can be implemented in ZnO microchannels, preferably in dark conditions, if subsequent intact DNA recovery and analysis is desired. Indeed, in this case, prolonged incubation of bacteria in microchannels with nanotextured ZnO walls is recommended in the dark, where direct (mechanical) bacteria killing contributes significantly to lysis, while ROS-mediated oxidative damage of released DNA is restricted. On the contrary, light conditions in microfluidic channels with nanostructured ZnO are preferable for applications where DNA inactivation is targeted, as in the example that follows. 

### 2.5. DNA Decontamination in Microchannels with Nanostructured ZnO Walls

Undesirable DNA often exists in trace amounts in commercial amplification kits as a contaminant originating from the extraction process of the enzyme content of the kit from bacterial cells. This contaminant DNA may interfere with the amplification result (false positive) when the amplification kit reagents are added to the target DNA for the amplification reaction to occur. An example is the *E. coli* DNA contamination often appearing in the recombinase polymerase amplification (RPA) kit, originating from the enzymes contained in these kits. Thus, irradiation of the enzyme with UV light is required in order to inactivate *E. coli* DNA [51,59] before the amplification reaction. To demonstrate DNA decontamination in the microfluidic device proposed herein, the contents of two RPA reagent tubes were diluted in 60 μL of NaCl 0.9% and the solution was injected in two microfluidic channels (with PMMA and ZnO walls, as in Figure 3a). After resting in the microchannels for 30 min, the solutions were collected and were subsequently used for PCR amplification together with an RPA reagent solution diluted by a factor of 10 (in NaCl 0.9%) and used directly in PCR. The gel electrophoresis image in Figure 7 shows the amplification of the ybbw gene of *E. coli* (at 210 base pairs), demonstrating contamination of the enzyme in the kit. In addition, the amplification signal from the nanostructured ZnO microchannel is about 10% of the intensity of the signal from the PMMA microchannel (and similar to the diluted by 10 RPA solution), indicating sufficient decontamination (by 90%) of the RPA kit after 30 min in the nanostructured ZnO microchannel presented herein under light conditions.

To sum up, the antibacterial properties demonstrated by nanostructured ZnO surfaces were transferred to the microfluidic channel device proposed herein, embedding the examined surfaces as microchannel walls to demonstrate the potential implementation of such a device for reagent-free bacterial cell lysis as part of a lab-on-a-chip accommodating bacterial lysis and downstream DNA-based bacterial analysis. Although the use of a simple microfluidic channel did not enhance bacterial cell lysis compared to its counterpart surface, an easy-to-fabricate-via-a-“green”-approach nanostructured ZnO-based microchannel device with a dual functionality under dark/light conditions was demonstrated. On one hand, the very high amount of intact DNA (as discussed in Section 2.4) released in the nanostructured ZnO microchannel under dark conditions proves this device to be appropriate for reagent-free bacterial cell lysis. On the other hand, the same device operated under light conditions is recommended for decontaminating biological samples from contaminant DNA, as was demonstrated in this section for RPA commercial kits contaminated with undesirable DNA.

## 3. Materials and Methods

### 3.1. Materials

Poly (methyl methacrylate) (PMMA) plates, 3 mm thick, were obtained from IRPEN (Barcelona, Spain) and cut in 1 × 1 cm^2^ or 1.5 × 1.5 cm^2^ square pieces, used as control samples for bacteria viability experiments on various substrates. Zinc (Zn) sheets (1-mm thick) of 99.9% purity were purchased from Klöckner and Co, Duisburg, Germany. Zn and PMMA substrates, 1.6 × 2.8 cm^2^ in size, were also used for patterning a microchannel on them by milling. For milling, a FEHLMANN Picomax 56 tool was used. Zn substrates were subsequently nanotextured, as it will be described below.

Poly (dimethyl siloxane) (PDMS) Sylgard 184 silicone elastomer kit was purchased from Dow Corning (Midland, MI, USA). All media and chemical reagents used for bacterial assays were purchased from PanReac AppliChem, while Luria-Bertani (LB) was prepared in house. For PCR, the KAPA2G Fast ReadyMix (KapaBiosystems, Wilmington, MA, USA) kit was used according to the supplier’s instructions. 

### 3.2. Surface Modification and Characterization 

Nanostructured zinc oxide (ZnO) substrates of 1 mm in thickness and 1 × 1 cm^2^ or 1.5 × 1.5 cm^2^ in size were prepared by implementing the hot water bath treatment method. More specifically, zinc samples were thoroughly cleaned with acetone and isopropyl alcohol using ultrasonic processing to remove organic contaminants prior to nanostructuring. Subsequently, glass beakers filled with deionized water were heated using a hotplate until the temperature of the water reached 95 °C. The temperature was fixed at this point and the zinc samples were immersed in the hot water. The immersion time of the samples used in this work was 24 h, and the surface nanostructures were examined afterwards. A Scanning Electron Microscope (SEM), the JSM 7401F Field Emission SEM from JEOL, was used to characterize the morphology of the tested nanostructured ZnO surfaces. 

### 3.3. Bacterial Viability on Various Substrates 

The Gram-negative bacteria, *Escherichia Coli* (strain, TOP 10 from INVITROGEN, Waltham, MA, USA), was grown overnight in Luria–Bertani (LB) medium at 37 °C on a shaking incubator (by LABTECH, Hopkinton, MA, USA) at 200 rpm. After the overnight incubation, the bacteria cells were resuspended into a fresh LB media and incubated for 2 h to obtain mid-exponentially-growing cells at the optical density (OD) of about 0.35–0.5. The optical density of *E. coli* at 600 nm was measured to assess the bacteria concentration in the culture medium. Then, the bacteria cells were harvested after 2 rounds of centrifugation (4000 rpm, at RT, 10 min) and resuspended in 0.9% NaCl, pH = 7, followed by sequential dilution to reach a final concentration range of ~10^6^ or ~10^8^ colony-forming units (CFU) mL^−1^ for subsequent experiments with the prepared surfaces. 

Bacterial viability was evaluated on 3 substrates (flat zinc, nanostructured ZnO, and flat PMMA as control) via colony counting, and the number of colony-forming units per mL of the original sample was calculated. Following the microplating assay, first, a volume (2 mL) of bacteria suspension was inoculated on the surfaces to be tested. The cell suspension was uniformly distributed on top of the surfaces, while the samples with a surface area of 1 × 1 or 1.5 × 1.5 cm^2^ were submerged in 12-well-plates. All samples were first sterilized by applying three steps: 70% ethanol to remove cell debris for 10 min and cleaning with double distilled water (ddh2O) and dry pure air of N_2_. The bacteria-inoculated surfaces were incubated at 37 °C for 1 h or 15 min at stationary conditions. After incubation, serial dilutions of the eluent of each tested surface were transferred to LB agar plates to assess the colony forming unit capacity of surviving cells using the microplating technique. Before the inoculation in agar plates, each diluted sample, in aliquots (500 μL), was mixed in vortex (uniform distribution of cells in solution), and this was followed by the inoculation of the 5 μL into agar plate. The procedure was repeated three times for each sample (three plates), and the same surface sample was performed in triplicates for statistical purposes. Then, the plates were incubated at 37 °C for 12 h. The viability of live bacteria appeared in colonies in LB agar plates. The killing rate per unit time and surface area was estimated, as follows: $$Killingrate(\mathrm{CFU}/(h\;{\mathrm{cm}}^{2}))=\frac{N_{control}-N_{tested}}{t(h)\;A({cm}^{2})}$$
where N_control_ is the number of cells (CFUs) in eluent sample on the control surface, N_tested_ is the number of cells in eluent on tested surfaces, t is the residence time, and A is the sample surface area. To determine whether bacteria cells were attached to the tested surfaces, 1 mL of 0.9% NaCl was used to wash the tested surfaces, and then the solution was incubated on agar plates to assess the colony forming unit capacity of surviving cells, as described above.

### 3.4. Bacterial Viability in Microchannels 

Incubation with *E. coli* suspensions was also performed in microchannels. A meandering microchannel 0.2 mm in depth, 2 mm in width, and 6.4 cm in total length, shown schematically in Figure 8, was patterned by means of CNC milling on PMMA and Zn substrates (prior to ZnO nanostructuring). The total microchannel volume was 26 μL, while the total microchannel surface was 2.8 cm^2^. On the zinc substrates and microchannel, ZnO nanostructures were formed after 24 h immersion in a hot water bath (the ZnO surface area in the microchannel was 1.54 cm^2^). A Poly (dimethyl siloxane) (PDMS) slab, 3 mm thick, was prepared for reversibly sealing the microchannel. PDMS was prepared by mixing the base (Sylgard 184 silicone elastomer kit, Dow Corning, Midland, MI, USA) with the curing agent at a volume ratio of 10:1, casting in the mold and baking at 80 °C for 2 h. An aluminum chip holder was used to press mechanically the PDMS cover on the PMMA or the ZnO-patterned substrates and prevent leaking. 

*E. coli* suspensions were incubated in the PMMA (control) and ZnO microchannels to estimate the killing rate of bacteria in the channel. *E. coli* suspension with a concentration of 10 ^8^ cells/mL was incubated in the microchannel at static conditions. The samples were manually loaded in the microchannels and remained in the incubator for 15 min at 37 °C. The colony counting method was performed after aspiration of 26 μL bacterial solution (equal to the total channel volume) from the outlet. During incubation, the microchannels were kept under light or in the dark in order to examine the effect of light on the antibacterial activity of microchannel wall materials.

### 3.5. DNA Measurements 

The DNA concentration (ng/μL) in samples of 2 μL supernatant from cell suspensions (10^8^ CFUs/mL), after incubation on surfaces and microchannels, was measured via a UV–VIS spectrophotometer (Nanodrop 1000, ThermoFisher Scientific, Paisley, UK).

The *E. coli*-specific ybbW gene sequence was amplified by PCR to confirm that DNA release in the cell lysate was a result of bacteria killing after exposure to modified surfaces. The PCR reaction was performed using the Fast Start^TM^ Master Mix polymerase (KAPA2G Fast ReadyMix, KapaBiosystems, Wilmington, MA, USA). The Master Mix kit contained a 2× condensed mixture of the Fast Start Taq DNA polymerase enzyme, the reaction buffer, and nucleotides. The PCR reagent total volume consisted of 13.4 µL of ddH2O, 25 μL of Fast Start Master, 1.5 μL of forward primer 10 pmol/μL (F primer), 1.5 μL of reverse primer 10 pmol/μL (R primer), and 1 μL of crude DNA-containing solution, obtained from the supernatant of *E. coli* cells after the antibacterial assay on various substrates and microchannel surfaces. The PCR reaction was performed in a Thermal Cycler (Τ-Personal 005-552 from Biometra). For comparison, 50 ng of genomic *E. coli* TOP10 DNA was used as a purified template. An amount of 1 mL of cell suspension from the well-plates containing the various substrates was transferred in aliquots and was centrifuged at 4000 rpm for 5 min to precipitate the DNA from the lysed cells and be suitable for PCR reagents in terms of DNA purity. In the same manner, 26 μL volume of cells from the microchannel outlet was aspirated after the antibacterial assay and was centrifuged at 8000 rpm for 10 s keeping debris of cells at the bottom. The thermocycler’s PCR protocol consisted of three steps: 10 s of denaturation at 95 °C; 10 s of annealing at 60 °C; and 10 s of extension at 72 °C, all of which were repeated 30 times. The PCR products were then visualized using an ultraviolet (UV) visualizer and loaded onto agarose gel (2%) stained with ethidium bromide (from AppliChem). The gel was imaged with GelAnalyzer 19.1 (free desktop app for 1D gel electrophoresis evaluation). For the electrophoresis kit on agarose gel, a DC voltage with 95 mA and a running time of 20 min were used.

In addition, ethidium bromide was used as an indicator of DNA presence in the crude cell lysate. Using a Spark^®^ (10M, TECAN) multimode microplate reader, the fluorescence intensity of EtBr following intercalation in the DNA released from the microchannels was quantified. A spectrofluorometer cuvette of each sample containing 180 μL of DI water and 30 μL of EtBr from a stock solution at a concentration of 0.4 mg/L was filled with a 20 μL aliquot of the bacterial cell lysate extracted from the microchannel. The Tecan-Spark spectrofluorometer measured the fluorescence intensity of EtBr at a wavelength of 595 nm. Two control samples were used to compare the measurements and determine the intensity, which is proportional to the DNA concentration: one contained deionized water, 0.9% NaCl solution and ethidium bromide solution; while the second reference sample only contained deionized water.

### 3.6. ROS Quantification

Dihydrorhodamine 123 (DHR123) is a neutral, non-fluorescent dye that is derived from rhodamine 123 (R123). In the presence of ROS, DHR123 undergoes oxidation and transforms into the fluorescent R123. In this work, we have used Hidex Sense Microplate Reader to measure the fluorescence. The fluorescent data recorded by the tool is the direct measure of the amount of ROS generated. The ROS is reported as relative ROS% by taking the control as reference. 

Nanostructured ZnO and flat PMMA samples were placed in 6-well plates in triplicates; 4 mL of deionized water was added to each well with the samples and to 3 wells without samples, which were used as control wells. In addition, 200 μL of 10 mM Dihydrorhodamine 123 (from Invitrogen) was added in each well. Two sets of experiments were carried out—one in light conditions and a second one in dark conditions. Reactive oxygen species (ROS) were measured at the beginning of the experiment as a reference point, and this was repeated after 0.5 h up to 5.5 h.

## 4. Conclusions

The present work leverages the antibacterial properties of nanostructured zinc oxide (ZnO) surfaces by incorporating them as walls in a simple-to-fabricate microchannel device for bacterial lysis. It also sheds light on the mechanisms responsible for *E. coli* bacteria killing in contact with zinc and nanostructured ZnO substrates and their respective microchannel surfaces, as compared to control PMMA surfaces. Prolonged bacterial interaction with relatively small (1 × 1 cm^2^) ZnO nanostructured surfaces contributes to bacteria killing (4.1 × 10^5^ CFU/cm^2^ h) partly via mechanical action. For short bacteria interaction (~15 min) with large nanostructured ZnO surfaces (1.5 × 1.5 cm^2^), indirect bacteria killing due to ROS formed from Zn and ZnO surfaces dominates the killing rates (7.8 × 10^7^ CFU/cm^2^ h). The remarkable killing rate of *E. coli* in nanostructured ZnO microchannels (1.9 × 10^6^ CFU/cm^2^ h), even after a 15 min incubation of the bacteria, paves the way for the implementation of nanostructured ZnO surfaces in both the disinfection of bacteria-containing solutions and the reagent-free lysis of bacteria injected in such easy-to-fabricate ZnO microchannels, following a “green” fabrication approach. This work also investigates the quality of DNA released as a result of bacteria killing. Due to the formation of ROS in the presence of nanostructured ZnO, significant inactivation of DNA (by a factor of 1.5) was observed for 15 min residence time of DNA in the ZnO microchannel. This may find application in cases where the presence of DNA in a sample is not desirable, as in contaminated commercial RPA amplification kits which were demonstrated for the first time in this work. Despite partial inactivation of the DNA in the ZnO microchannel under light, free and intact bacterial DNA was detected through PCR amplification of the whole cell lysate as well as through fluorescence of EtBr intercalated in free DNA released, in the dark, only in nanostructured ZnO-based microchannels. Thus, microfluidic devices with nanostructured ZnO microchannel surfaces, operating in the dark (for reduced DNA inactivation) and combining high bacteria killing rates with the release of free, intact bacterial DNA, can pave the way for their implementation as efficient, reagent-free bacterial lysis modules for subsequent DNA analysis in more integrated lab-on-a-chip devices. The demonstrated potential of the nanostructured ZnO microchannel device for tailored dual functionality in light/dark conditions is the main novel contribution of the present work.

## Figures and Tables

**Figure 1 antibiotics-12-01276-f001:**
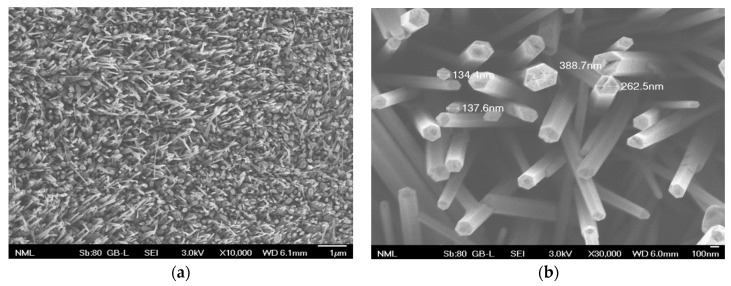
SEM images (at different magnifications, ×10,000 in (**a**) and ×30,000 in (**b**)) of ZnO nanostructured substrates obtained after 24 h immersion of Zn in hot water bath. ZnO nanorods are shown uniformly distributed on the surface.

**Figure 2 antibiotics-12-01276-f002:**
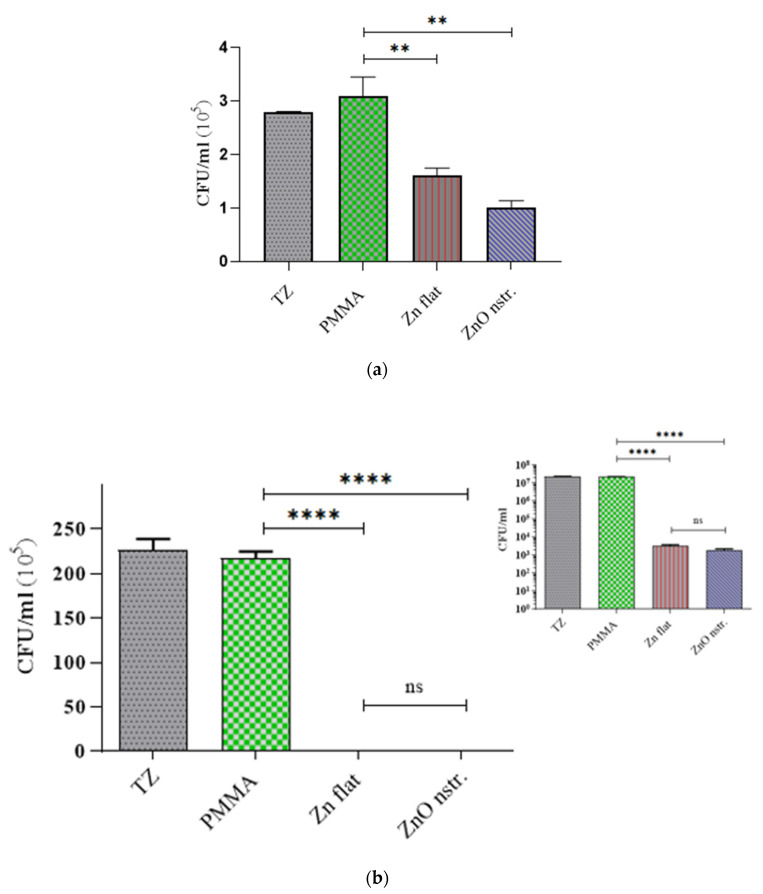
*E. coli* viability on nanostructured ZnO, flat Zn, and PMMA substrates (**a**) 1 × 1 cm^2^ and (**b**) 1.5 × 1.5 cm^2^ in size after 1 h and 15 min incubation time, respectively. The bar values represent the mean value of two independent experiments (two substrate surfaces); error bars indicate the standard deviation. From each substrate surface, bacterial suspension was collected and was inoculated to agar plates in triplicate. TZ is the bacterial concentration at time zero. For clarity, both the linear and logarithmic (inset) *y*-axis is used in (**b**). ‘**’ means that the *p* value < 0.005, while ‘****’ means *p* < 0.00005; therefore, the measured values are significantly different. The measured values for flat Zn and nanostructured ZnO are not significantly different (ns).

**Figure 3 antibiotics-12-01276-f003:**
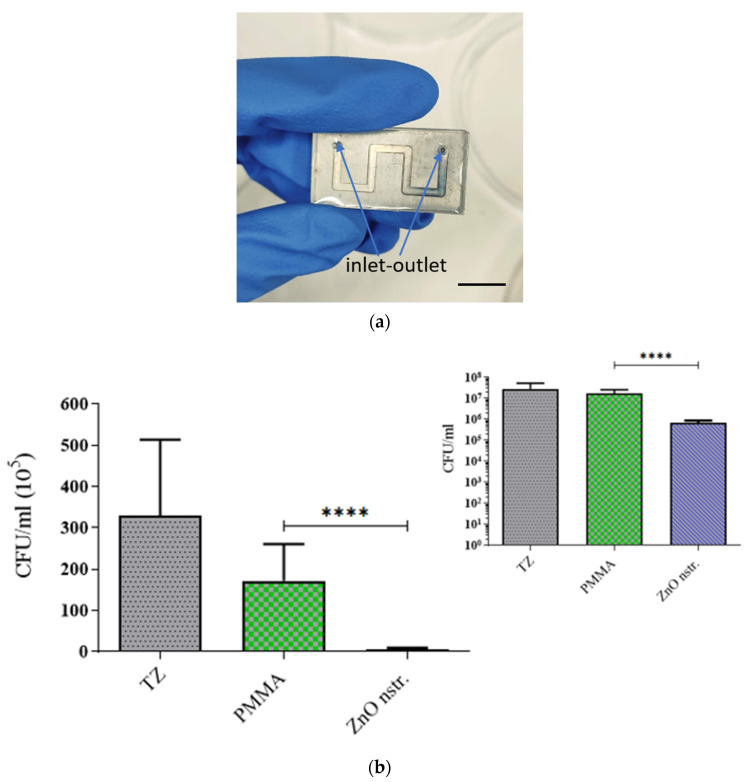
(**a**) A microchannel formed on a Zn substrate, nanostructured, and reversibly sealed with a PDMS cover bearing an inlet and outlet for bacteria solution injection and collection, respectively; bar size 1 cm. (**b**) *E. coli* viability on nanostructured ZnO and flat PMMA microchannel walls after 15 min incubation time. For clarity, both the linear and logarithmic (inset) *y*-axis is used. The bar values represent the mean value of two independent experiments (two microchannel devices); error bars indicate the standard deviation. ‘****’ means that the *p* value is less than 0.00005; therefore, the measured values are significantly different.

**Figure 4 antibiotics-12-01276-f004:**
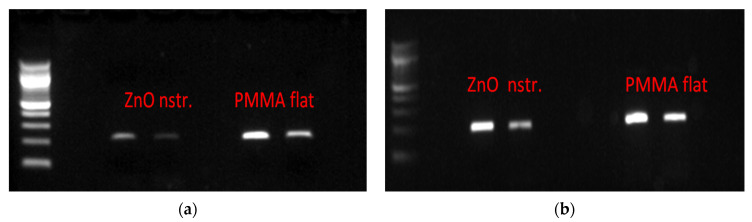
Images from gel electrophoresis of PCR amplified ybbw gene of (**a**) *E. coli* lysed in ZnO and PMMA microchannels. Lane 1: DNA ladder (Nippon); Lanes 2 and 3: lysate from ZnO microchannel (diluted 1:30, 1:60); Lanes 4 and 5: lysate from PMMA (control) microchannel (diluted 1:30, 1:60). (**b**) purified *E. coli* DNA after residing for 15 min in the microchannels. Lane 1: DNA ladder (NEB); Lanes 2 and 3: pure DNA from ZnO microchannel (diluted 1:30, 1:60); Lanes 4 and 5: pure DNA from PMMA (control) microchannel (diluted 1:30, 1:60).

**Figure 5 antibiotics-12-01276-f005:**
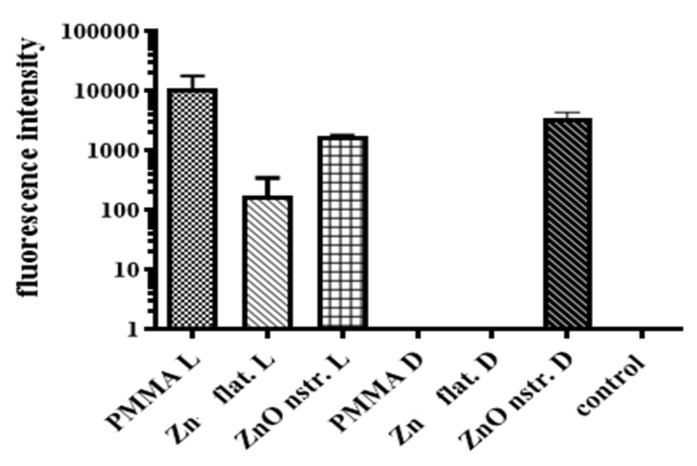
Fluorescence intensity for EtBr intercalated into free DNA from lysed *E. coli* in nanostructured ZnO, flat Zn, and PMMA (control) microchannels under light (L) and dark (D) conditions.

**Figure 6 antibiotics-12-01276-f006:**
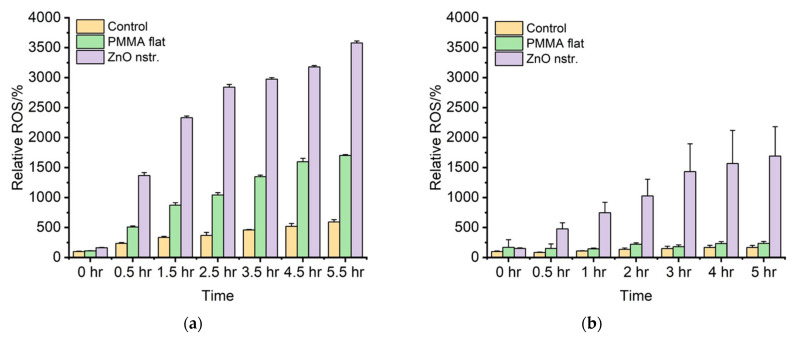
Percentage of reactive oxygen species produced in nanostructured ZnO and flat PMMA surfaces under (**a**) light and (**b**) dark conditions. The error bars represent the standard deviation of three measurements for each type of sample.

**Figure 7 antibiotics-12-01276-f007:**
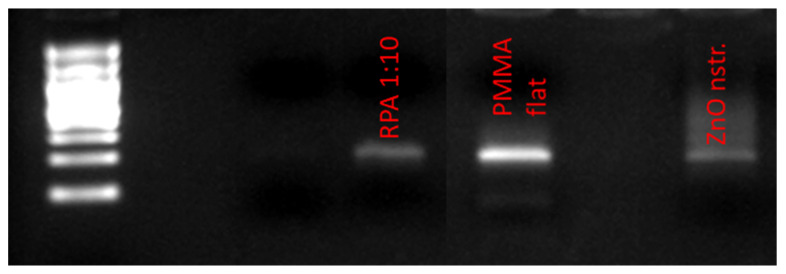
Image from gel electrophoresis of PCR amplified contents of a commercial RPA reagent kit. Lane 1: DNA ladder (Nippon); Lane 2: amplified product directly from RPA kit (diluted 1:10); Lane 3: amplified product from PMMA (control) microchannel (residence time: 30 min); Lane 4: amplified product from ZnO microchannel (residence time: 30 min).

**Figure 8 antibiotics-12-01276-f008:**
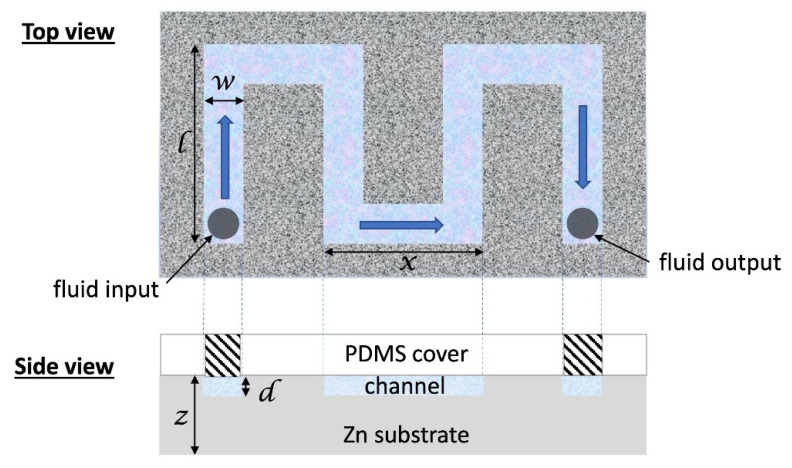
Schematic of the microchannel device fabricated on a Zn substrate followed by surface nanostructuring. The geometrical dimensions are as follows: microchannel width and depth, *w* = 2 mm and *d* = 0.2 mm, respectively; *l* = 10 mm, *x* = 8 mm (total microchannel length 64 mm). The microchannel was sealed with a PDMS cover.

**Table 1 antibiotics-12-01276-t001:** Killing rates for *E. coli* on Zn and nanostructured ZnO substrates and microchannel surfaces.

	Surface Area (cm^2^)	Initial Concentration (CFU/mL)	Incubation Time (h)	Killing Rate (CFU/cm^2^ h)
Zn flat substrates	1 × 1	~10^6^	1	(3.0 ± 0.5) ×10 ^5^
	1.5 × 1.5 (2.25)	~10^8^	0.25	(7.8 ± 1.0) × 10^7^
ZnO nanostructured substrates	1 × 1	~10^6^	1	(4.1 ± 0.7) × 10^5^
	1.5 × 1.5 (2.25)	~10^8^	0.25	(7.8 ± 1.0) × 10^7^
ZnO nanostructured microchannel	1.54	~10^8^	0.25	(1.9 ± 1.0) × 10^6^

**Table 2 antibiotics-12-01276-t002:** DNA concentration estimated by means of UV absorption.

	DNA from Lysed Bacteria after 15 min Residence (ng/μL)	Purified DNA before Introduction in Microchannels (ng/μL)	Purified DNA after 15 min Residence (ng/μL)
Flat PMMA microchannel (control)	21	52	55
Nanostructured ZnO microchannel	6	52	25

## Data Availability

The data presented in this study are available on request from the corresponding author. The data are not publicly available at the time of acceptance, due to the current lack of a data repository.

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
