# Peer review of "A High-Performance Antibacterial Nanostructured ZnO Microfluidic Device for Controlled Bacterial Lysis and DNA Release"

_antibiotics, 2023, doi:10.3390/antibiotics12081276_

Round 1

Reviewer 1 Report

See attachment

Author Response

Please find attached the reply to Reviewer 1.

Reviewer 2 Report

Reference style and units should be checked. 

This paper mentions bacterial lysis with DNA release on flat PMMA and ZnO nanostructured surfaces. The reviewer could not well understand the importance of this study for the DNA release control, and thought that some important experimental data were missing. The presentation of the results is not clear for the potential readers of this manuscript. The reviewer recommends the authors to revise this paper according to following comments.

<Major comments>

#1: In the fourth paragraph of introduction, the author should discuss why use of nanostructured ZnO is suitable for bacterial cell lysis? The authors only mentioned the use of the ZnO microstructures for antibacterial activity. The activity is different with bacterial lysis.

#2: Table 1, are the killing rates with significant difference?

#3: Figure 5, Is the fluorescent intensity correlated with the intercalated DNA inside the microfluidic channels? The relationship between the DNA content and the EtBr fluorescent intensity should be presented.

#4: Figure 6, the presentation of (a) and (b) should be unified. What do the error bars show?

#5: Zn flat surface data are necessary for supporting authors’ claim. In Figure 4, 5, 6, and 7, the experimental results on Zn flat surface are mandatory. The reviewer confused and cannot be convinced with the claim due to lack of the results.

#6: What results are provided using the microfluidic channels? The authors may not use the microfluidic channels to show data in Figure 2 and Table 1. Is this correct? Anyway, the use of microfluidic channel is not clear throughout the manuscript. Moreover, what improved in the use of the microfluidic channel for the cell lysis and the undesirable DNA existence? The reviewer had this question, because the relative ROS and PCR amplification data are missing in the use of Zn-flat surface microfluidic channels!

#7: Can the authors show the difference of PCR amplified contents between the light and dark conditions? The reviewer cannot agree with the claim “tailored dual functionality in light/dark conditions” based on the present results.

#8: The reviewer could not understand why the removal of DNA contamination after the bacterial lysis is required in practical application of the bacterial lysis and DNA analyses. The reason may be reducing undesired PCR byproducts from the bacterial lycete. If the authors recognize the necessity of reducing the PCR byproducts, discuss this point anywhere with references. If the scenario may be true, increased ROS may decrease the target DNAs. Is this a problem or not?

#9: How flat Zn and PMMA surface? Show the SEM images of the Zn and PMMA surface anywhere.

#10: The section “Results and discussion” should be placed after “Materials and methods”. This order is quite strange.

Author Response

Please find attached the response to Reviewer 2.

Round 2

Reviewer 2 Report

The reviewer thinks that the manuscript writing improved in the revised version. However, the reviewer requests to reflect following three points to the revised manuscript.

* In Figure 5, a standard curve of EtBr Fluorescence intensity (FI) with DNA concentration (or contents) should be disclosed anywhere (in Supporting Information also acceptable). The linearity between FI and DNA concentration should be evaluated in each experimental system. FI saturation may occur in high FI value, and then the linear relationship cannot be found. If the authors have already shown this calibration curve in other paper, the refer to the previous paper is also acceptable.

* Mention the meanings of ‘**’ and ‘****’ (maybe P values of t-test) in Figures 2 and 3.

* If possible, the answer of Comment #6 should be reflected to the revised manuscript. Some readers may consider that the importance to use the microfluidic channel cannot be found in this revised manuscript.

Author Response

Response file uploaded
